Interactive bioacoustic playback as a tool for detecting and exploring nonhuman intelligence: “conversing” with an Alaskan humpback whale

McCowan Brenda 1 bjmccowan@ucdavis.edu
http://orcid.org/0000-0001-5112-1329 Hubbard Josephine 2
Walker Lisa 3
Sharpe Fred 4
Frediani Jodi 5
Doyle Laurance 6
1 SVM: Population Health and Reproduction, University of California, Davis , Davis, California , United States
2 Animal Behavior Graduate Group, University of California, Davis , Davis, California , United States
3 Grooved Whale Project , Vancouver, British Columbia , Canada
4 Alaska Whale Foundation , Petersburg, Alaska , United States
5 Jodi Frediani Photography , Santa Cruz, California , United States
6 SETI Institute , Mountain View, California , United States
Oberst Sebastian
Electronic publication date: 2023 Nov 29
Publication date: 2023
Volume: 11
Electronic Location ID: e16349
Received 2023 Feb 9; Accepted 2023 Oct 4
Copyright: © 2023 McCowan et al.
Copyright year: 2023
Copyright holder: McCowan et al.
License: This is an open access article distributed under the terms of the Creative Commons Attribution License, which permits unrestricted use, distribution, reproduction and adaptation in any medium and for any purpose provided that it is properly attributed. For attribution, the original author(s), title, publication source (PeerJ) and either DOI or URL of the article must be cited.
License URL: https://creativecommons.org/licenses/by/4.0/

Keywords: Communication, Bioacoustic playback, Turn-taking, Vocal matching, Humpback whale

Funding: Templeton World Charity Foundation Diverse Intelligences TWCF0440 This research was supported by Templeton World Charity Foundation Diverse Intelligences grant TWCF0440 (Laurance Doyle). The funders had no role in study design, data collection and analysis, decision to publish, or preparation of the manuscript.

==============================
Here we report on a rare and opportunistic acoustic turn-taking with an adult female humpback whale, known as Twain, in Southeast Alaska. Post hoc acoustic and statistical analyses of a 20-min acoustic exchange between the broadcast of a recorded contact call, known as a ‘whup/throp’, with call responses by Twain revealed an intentional human-whale acoustic (and behavioral) interaction. Our results show that Twain participated both physically and acoustically in three phases of interaction (Phase 1: Engagement, Phase 2: Agitation, Phase 3: Disengagement), independently determined by blind observers reporting on surface behavior and respiratory activity of the interacting whale. A close examination of both changes to the latency between Twain’s calls and the temporal matching to the latency of the exemplar across phases indicated that Twain was actively engaged in the exchange during Phase 1 (Engagement), less so during Phase 2 (Agitation), and disengaged during Phase 3 (Disengagement). These results, while preliminary, point to several key considerations for effective playback design, namely the importance of salient, dynamic and adaptive playbacks, that should be utilized in experimentation with whales and other interactive nonhuman species.

Introduction

The study of nonhuman animal signals challenges and informs our search for nonhuman intelligence with the ultimate hope of deepening the human-animal relationship. The diversity of nonhuman intelligence that exists on Earth has been revealed in a multitude of observational and experimental studies over the past several decades (Vonk, 2021). This diversity is observed from octopus to ravens to elephants to whales, thus manifesting itself in a variety of both terrestrial and non-terrestrial environments. Indeed, one needs look no further for a non-terrestrial form of intelligence than in the remarkable acoustic, behavioral and social versatility of the humpback whale (Megaptera novaeangliae) (Payne & McVay, 1971; Fournet & Szabo, 2013; Fournet, Szabo & Mellinger, 2015).

Whales are ancestrally remote to humans and resumed life in the oceans 60 million years ago. Here humpback and other whales found themselves in an ocean soundscape capable of distant and widespread transmission of both low and high frequency sounds (Payne & Webb, 1971). The brain of the humpback whale possesses a large auditory cortex and clusters of spindle neurons that are presumably linked to their altruistic behavior (Hof & Van der Gucht, 2007; Pitman et al., 2017). Their global recovery from whaling hints at remarkable survival skills. The humpback’s superlative acoustic prowess is likely underwritten by impressive mental abilities. The whales acoustic achievements include songs that are lengthy, rhythmic and constantly evolving (Payne & McVay, 1971; Garland et al., 2011). Structural parallels between human speech and humpback whale song (among vocalizations of other nonhuman species) include the use of a similar frequency band and are comprised of both tonal (voiced) and broadband (unvoiced) elements, and vocalizations of variable duration punctuated by silence (Pace et al., 2010). Their vocal repertoire also includes a panoply of less-studied social sounds (non-song sounds), including over 40 unique social calls (Thompson, Cummings & Ha, 1986; Cerchio & Dahlheim, 2001; Dunlop et al., 2007; Fournet & Szabo, 2013; Fournet, Szabo & Mellinger, 2015; Fournet et al., 2018). These calls are produced in bouts whose variability increases with social complexity and larger group sizes (Tyack, 1983; Silber, 1986; Cusano et al., 2020). A subclass of these calls exhibits stability across generations and ocean basins, which is often considered a prerequisite for the evolution of complex communication (Bradbury & Vehrencamp, 2011; Rekdahl et al., 2013; Wild & Gabriele, 2014). At least one of these calls is thought to serve as a contact call known as the “whup” or “throp” call (Wild & Gabriele, 2014; Fournet & Szabo, 2013; Fournet, Szabo & Mellinger, 2015; Fournet et al., 2018). This diversity of signals, which can be a key indication of behavioral flexibility, represents an excellent opportunity to characterize and interact with a potential nonhuman intelligence.

Communication is a fundamental behavior of all living beings that facilitates the exchange of information among signalers and receivers and is dynamically comprised of both reciprocal and emergent processes (Wiley, 1983; Bradbury & Vehrencamp, 2011). Researchers of communication, specifically in nonhuman animals, have utilized a series of tools to decipher both the structure and function of these communication systems to gain greater insight into the diversity, complexity and meanings of signals animals use to navigate their social environments (Wiley, 1983; Kershenbaum et al., 2016; Fischer, Noser & Hammerschmidt, 2013; Reiss, McCowan & Marino, 1997; McCowan, Hanser & Doyle, 1999, 2002). While most communication systems are multimodal in nature, a wealth of research has specifically focused on the acoustic communication of a multitude of animal species (Wiley, 1983; Bradbury & Vehrencamp, 2011; Kershenbaum et al., 2016) as a means toward a greater understanding of the evolution of vocal communication (including human language) as well as a tool for enhancing welfare and conservation efforts McCowan & Rommeck (2006). The tools involved in this pursuit include the bioacoustic recording of signals or calls, noting their behavioral contexts, and analyzing their acoustic structure (comprised of spectral and temporal components) to decipher call meaning (Kershenbaum et al., 2016; Fischer, Noser & Hammerschmidt, 2013). A serious challenge to these traditional types of observational approaches however is that context does not always predict the signal to be used nor conversely does the signal precisely predict its context; that is one signal does not always equal one context. Rather, it depends on the signal’s function; for example, alarm calls generally tend to be stereotypical in acoustic structure and are emitted conservatively in very specific contexts (e.g., presence of a predator), while other less constrained calls, such as a general nondescript category known as “contact calls”, are more variable in acoustic structure and emitted more ubiquitously in multiple contexts (Fischer, Noser & Hammerschmidt, 2013; McCowan, Hanser & Doyle, 1999; Ferrer-i Cancho & McCowan, 2009). Additionally, calls can occur in long and dynamically changing sequences such as in bird and whale song, further complicating the deciphering of call meaning not only in different social but also acoustic contexts (like in human language) (Payne & McVay, 1971; Rekdahl et al., 2015; Kershenbaum et al., 2016; Miksis-Olds et al., 2008; McCowan, Hanser & Doyle, 1999, 2002).

In addition to variation in context, many forms of communication involve turn-taking, which is defined as the alternation of information exchange resulting in observable temporal regularities between two or more interactants (Pika et al., 2018). Turn-taking prevents interruptions and the inefficiency of multitasking (Pika et al., 2018). Turn-taking aligns along a spectrum from simple to complex and its sophistication provides insight into a given species communicative prowess and perhaps intelligence. Simple forms of turn taking include insect vibration, percussion, stridulation, and bioluminescence (grasshoppers, cicadas, and fireflies), while more complex examples are observed in primates with gaze, gesticulation, and antiphonal calling. In humans, spoken language underwrites turn taking cadence and has been proposed as a potentially cognitively demanding process. This is due to the rapid turn response rate ( ≈200 ms) compared to average language processing speeds ( ≈600 ms), potentially allowing receivers to begin crafting their response before fully receiving a signal (Levinson & Torreira, 2015). As a result, speakers construct their turns out of units whose structure allows the next speaker to anticipate their completion (Pika et al., 2018). In nonhuman animals, the turn response rate is much slower (ex. marmosets: 3–5 s), likely stemming from differences in ‘units of perception’ where animals of different sizes experience the world at different rates (Healy et al., 2013). In animals, the turn response rate can also be tied to several behavioral or social factors such as territoriality (manakins: Maynard et al., 2012; blue tits: Poesel & Dabelsteen, 2005), competition (barn owl siblings: Ducouret et al., 2018), or mating behavior (Kipper et al., 2015). Vocal turn-taking requires a certain level of coordination between partners to avoid overlapping signals. As a result, turn-taking has been shown to coincide with social organization (Pougnault et al., 2022) and has been thought to influence social coordination (Starlings: Henry et al., 2015) or enhance social bonds through cooperation (meerkats: Demartsev et al., 2018). For example, in species that are highly monogamous and develop strong bonds between mating pairs it has been shown that the longer the duration of the pair bond, the more similar their calling behavior will become (i.e., their signals converge) (titi monkeys: Clink, Lau & Bales, 2019; zebra finches: D’Amelio, Trost & ter Maat, 2017). In fact, aspects of vocal synchrony during vocal exchanges (such as matching the type or timing of a signal or inter-call intervals between signals) have been shown to correlate with acute changes in arousal, valence, or potential engagement between participants (Briefer, 2012). Synchrony, in its purest form, is a perfect (or close) matching of temporal coordination (Ravignani, 2015) and one subclass, antisynchrony consists of perfect (or close) alternation, the latter of which is more readily found in the turn-taking observed in human and nonhuman communicative interactions (Pika et al., 2018; Mondémé, 2022; Levinson & Torreira, 2015). Thus, the temporal dimension of time matching can be an essential element to a conversation’s structure. For example, inter-call intervals have been found to be significantly shorter during vocal exchanges than between solo calls (dolphins: Nakahara & Miyazaki, 2010; Japanese macaques: Katsu et al., 2019). Metrical structure has also been found in the songs of humpback whales suggesting that temporal organization and coordination is not just salient but important to this species (Handel, Todd & Zoidis, 2009).

While many studies on turn-taking in humans or animals focus on intraspecies communication (Pika et al., 2018), very few focus on interspecies partners. However, these forms of communication have great potential to provide insights to the evolution of signaling behavior and the role of flexibility in producing acoustic signals. Intraspecies communication has likely evolved in response to direct survival needs and fitness benefits of sharing information with conspecifics. Species who can extend those communication techniques to interspecies interactions may show particular promise in responding to foreign signals in their environment, as a result of a rapidly changing world climate and other human-related disturbances. Such interspecies communication can include indirect eavesdropping on other species’ signals for identifying resources or avoiding predators (Magrath et al., 2015) but can also explicitly be designed to directly elicit responses from heterospecifics including from and by humans. When making predictions regarding interspecies communication patterns it is reasonable to expect that organisms with closer phylogenetic relationships may share mutual entrainment processes and intelligibility. However, it is also possible that with the appropriate selection pressures that these communicative capacities may also have evolved independently in more remote taxa. Indeed, studies on interspecies communication can allow for direct inquiries into shared communicative structure between species and has the potential to reveal “unplummed” sophistication and possibly intelligent behavior from nonhuman communicative partners. Studies on human-animal communication are particularly promising due to their potential to elicit a unique response from the subject outside of their species-specific, contextualized behavior patterns (Mondémé, 2022).

Finally, as one approach that includes an unusual type of “interspecies” communication, researchers have long explored the function and meaning of nonhuman animal signals using acoustic playback experiments (McGregor et al., 1992). These studies are designed to actively probe nonhuman communication systems beyond traditional observations to gain further understanding of signal meaning. Historically, the design and interpretation of these studies have been hotly debated, including concerns of accuracy and salience of acoustic stimuli, use of appropriate controls, pseudoreplication of playback sets, and reproducibility of methods and results (Fischer, Noser & Hammerschmidt, 2013; King, 2015; King & McGregor, 2016; Butkowski et al., 2011; Powell & Rosenthal, 2017; Pika et al., 2018; McGregor, 2000; De Rosa, Castro & Marsland, 2022; Deecke, 2007). One area of less focused concern however is the overall utility of acoustic playback in general, and especially passive acoustic playback, the primary mode in which acoustic playback has been conducted. Passive acoustic playback experiments take fixed stimuli of interest and broadcast them at randomized, predetermined intervals within the animal’s natural range to measure their behavioral and/or vocal response. While traditional passive acoustic playback has significantly contributed to our understanding of animal communication (Fischer, Noser & Hammerschmidt, 2013; King, 2015; King & McGregor, 2016; Butkowski et al., 2011; Powell & Rosenthal, 2017; Pika et al., 2018; McGregor, 2000; De Rosa, Castro & Marsland, 2022), it also has significant limitations. For example, in such experiments, concern arises over the composition and size of the audience targeted as well as the types of predictions one makes about appropriate responses, whether physical or vocal, to such playbacks. Indeed, if the response is predicted to be of a vocal nature, one needs to be careful about concluding that a simple increase in calling rate at the population level during playbacks to a large group of subjects is indicative of a contingent response since, in this context, the causal relationship between playback and response is unclear at best. Similarly, physical approaches to playback stimuli could be due to its function (as a contact call for example) or merely because the stimulus appears novel (or just novel enough) to the subject(s). In addition, such passive experiments are considered non-interactive because the playback stimuli are not adaptively modified to match or probe the varying vocal or physical responses by the subjects (King, 2015; King & McGregor, 2016). Yet, substantial evidence indicates that the information conveyed in a signal may only be salient (or more salient) when presented interactively or dynamically to (thus engaging) the subjects under study (King, 2015; King & McGregor, 2016).

Given this perspective, the goal of this report is to (1) describe a novel interaction, including a matched acoustic exchange via bioacoustic playback, with an individually identified humpback whale (known as Twain) in Southeast Alaska and (2) discuss how this particular interaction, and its successes and limitations, might inform future approaches in designing experimental playback studies that can more directly engage subjects and thus aid in the detection and exploration of intelligence in nonhuman species. Indeed, fostering an approach that permits more dynamic adaptive playback of other species’ vocalizations may give us great insight into the degree of behavioral flexibility and adaptability of nonhuman communications systems, a key element of intelligent behavior.

Methods

Portions of this text were previously published as part of a preprint (McCowan et al., 2023). On August 18, 2021, as part of collecting a diverse pool of call exemplars (Deecke, 2007), we recorded a high quality whup call (see Supplemental Material for spectrograms of the calls recorded) from a dispersed aggregation of nine humpback whales that were sighted in the general vicinity of our research vessel, Glacier Seal, which was drifting with engines off near Five Fingers in Frederick Sound, Alaska. This high-quality exemplar was used in a playback trial on the following day, August 19, 2021 approximately three nautical miles from the previous day’s location in which we conducted a controlled experiment with 20 min. pre-, during and post- playback periods (see Fig. 1) with a whale, subsequently identified via Happywhale (www.happywhale.com) as a 38+ year-old adult female known as Twain, who remained within 100 m of our research vessel during the entire playback period (see Fig. 2). The upper deck of the research vessel had four blind observers with a 360 ∘ view continuously collecting photos, video and notes on Twain’s behavior during the exchange, all of whom were blind to the playback paradigm. An additional still photographer (also blind to the paradigm) collected images on the lower deck. During the pre- and post-playback periods, four additional whales were observed via binoculars and telephoto lens between 200–1,000 m of the vessel and another eight whales were sited beyond 1,000 m from the vessel. Other than Twain, no other whales were present within 200 m of the vessel during the 20-min playback period. The playback apparatus, deployed from the lower deck of the Glacier Seal, consisted of a single underwater Lubell speaker (Lubell LL916) emitting the exemplar at approximately 6 m in depth, deployed on the port side of the bow, and two hydrophones (Cetacean Research Technology SQ26-01), deployed on the port and starboard side of the bow, approximately 3 m apart horizontally and at depths of approximately 3 and 6 m respectively. These hydrophones recorded the entire playback trial and exchange event on a MacBook Air computer using Audacity software (http://audacityteam.org/) and Zoom H1n Portable Recorder (see schematic in Fig. 2 and Supplemental Materials for full specifications of the equipment used and a detailed timeline of the encounter). Using a second Macbook Air computer, a single exemplar recorded the previous day was played back at first randomly (first three exemplars prior to whale’s initial response) and then dynamically adjusted based upon the calling behavior of the whale. These playbacks were therefore interspersed with the calls of the whale surfacing near our vessel at approximately 100 m of the playback apparatus and the two experimenters conducting the playback were blind to the whale’s surface behavior. Most of Twain’s recorded whup calls were of high amplitude (ranging from −21 to −43 dB). The overall site tenacity of this individual during the 20-min exchange as evidenced by seven blows/surfacings (which allowed us to periodically mark her position/distance/direction from the boat- see Fig. 2), followed by the calls dropping significantly in amplitude as she departed (GLM: β = −9.28, p = 0.0001), indicates with almost 100% certainty that Twain was the origin of these calls. As Fig. 1 shows, no vocalizations were recorded during the pre- or post-playback periods, only during the playback period. Three exemplars were broadcast prior to Twain’s first response and we ceased playback of exemplars prior to Twain’s 33rd call; thus, the final three calls by Twain were not in exchange with the playback exemplar after which time she ceased calling (see Fig. 3A). In all, there was a total of 38 whup exemplars broadcast and 36 Twain whup calls recorded during the 20-min playback period, during which time Twain performed seven blows/surfacings and observers captured three positive fluke IDs (see Supplemental Materials). All field investigations were conducted under National Marine Fisheries Service Research Permit No. 19703 to Fred Sharpe.

Figure 1 Representative spectrograms of (A) baseline control (pre) period, (B) experimental playback (during) period, and (C) follow-up control (post) period. No whup calls were recorded during pre- and post- periods. (D) Example spectrograms and waveforms of the whup exemplar and Twain’s whup call. Note exchange between E (exemplar) and T (Twain) in B.

Figure 2 Twain’s surface tracking and behavior during the playback phase of the trial (see Fig. 1B).

Whale image with text indicates position and behavior of Twain during playback trial. Boat image represents the R/V Glacier Seal. Hydrophones and speaker placement indicated. Fluke ID shot of Twain during the interaction.

Figure 3 Latencies between Twain’s calls.

(A) As a time series showing in red the latencies where Twain had surfaced/blew prior to vocalizing on Day 2 (playback) (blows 1 and 2 occurred just before the exchange began—see Fig. 2). Orange dotted lines indicate the range of latencies between whup calls on Day 1 (control). (B) Negative binomial regression on latencies between Twain’s calls during each of Day 2’s phases and the calls recorded on Day 1. (C) Latencies between the calls recorded on Day 1 (control) in comparison to Twain’s calls on Day 2 (playback). (D) Latencies between Twain’s calls when a counter call (direct exchange) vs not a counter call across the entire time series. (E) Randomizations (N = 1,000) of regression conducted in (D) indicating that the significant difference found between the latencies of Twain’s calls in counter calling vs. non-counter calling was not spurious.

To examine whether the acoustic sequence recorded during the broadcast of the exemplar and the whup vocalizations of Twain could be considered a causal human-animal interaction or exchange, we conducted a number of post hoc analyses on aspects of call structure of Twain’s whup calls recorded during this event. We considered Twain’s calls either to be counter calls or not, with counter calls defined as calls that occurred following an exemplar without any other intervening behaviors (surfacing/blows). Those not considered counter calls were Twain’s calls or other intervening behaviors, such as respiratory blows, produced in succession (see Supplemental Material for calls identified as counter-calls). The rationale for not including calls followed by intervening respiratory behaviors was because the act of breathing at the surface likely disrupted the ease of calling. We also categorized the entire interaction into three separate phases based on Twain’s behavior recorded by the blind observers. The independent behaviors used to define phases were surface blows and orientation to/distance from the research vessel. Phase 1 was considered “Engagement” as this was the onset of the exchange after three exemplars were broadcast (interspersed with two cryptic or neutral blows). Phase 2 was considered “Agitation” as its onset and progression included three wheezy or reverse forced surface blows. These types of blows are considered to indicate excitement/agitation in humpback whales (Thompson, Cummings & Ha, 1986; Watkins, 1977). Phase 3 was considered “Disengagement” as Twain was oriented away while increasing her distance from the research vessel (with two neutral blows). For an additional control and comparison to calls produced outside the playback trial context, we also included the latencies between whup calls from a short sequence of calls (N = 6) passively recorded from a small group of whales on August 18, 2021, the day before the playback trial (termed “Day 1”), the group membership of which included Twain and eight other whales of which Twain was the only individual common to both days (with the second day, the playback day, termed “Day 2“). It is important to note that we could not determine if the calls recorded were made by the same whale or were part of an exchange between two or more whales. Thus Day 1 serving as a ”control” day was only a control in the sense that we were not conducting a playback trial with a similarly sized group of whales in a similar location that included Twain.

Twain’s whup calls on Day 2 and the whup calls recorded on Day 1 were acoustically analyzed for both spectral and temporal features using Raven Pro 1.6.4 (https://ravensoundsoftware.com). After investigation of various spectral measures (e.g., minimum frequency, maximum frequency, center frequency, peak frequency contour), we limited our analysis to one variable of interest: the inter-call interval (or latency between calls as measured by the time difference between the preceding call’s offset and the subsequent call’s onset). This was primarily due to two reasons: (1) unknown changes in the orientation and distance of Twain from the hydrophone array (both of which can significantly change the spectral representation of even identical calls) and (2) the limitation in the use of a single whup exemplar that could only be manipulated via timing of broadcast in response to Twain’s calls. The rationale for this lone metric was that it potentially represents two fundamental aspects of emotional content of calls: arousal and valence (Briefer, 2012), which could be used to evaluate the motivational state of Twain and thus the causal nature of this communicative exchange.

In an additional analysis, we also examined whether the matching of Twain’s calls’ latencies to that of the exemplars (and exemplars to Twain’s calls) occurred (that is, the matching of response times) and how matching differed across the phases as defined by the behavioral context. Phase 3 had only one pair to match and thus was excluded from this analysis. To standardize across the absolute duration of the latency, ratios for each Exemplar-Twain (E-T) pair were calculated (N = 27 pairs) and conversely each Twain-Exemplar pair (N = 29 pairs). These ratios were used to develop two metrics of discrepancy. One was a categorical metric of “match or not”, where a perfect match would be a 50:50 ratio, in latency between pairs. Our criterion for designating a “match” was any ratio that occurred between 41:59 and 50:50 (see Supplemental Materials). To avoid any bias in defining thresholds for categorizing a match, the other measure was the absolute difference in the ratio between each pair as a second metric of discrepancy in latency matching. These metrics were included because such matching represents a type of behavioral synchronization, which is known to be an important feature in human and nonhuman animal communication (Wood et al., 2021; Bowling, Herbst & Fitch, 2013; Oesch, 2019; Imel et al., 2014; Lord et al., 2015; Xiao et al., 2013; Ravignani, 2018; Greenfield et al., 2021; Patel et al., 2009; King & McGregor, 2016; Hausberger et al., 2020; Demartsev et al., 2023; Herzing, 2015).

All metrics were subjected to GLM regression analysis where count outcomes were analyzed using negative binomial regression and binary outcomes using logistic regression in Stata 15 (StataCorp, 2017) or R Studio 7.2022 (RStudio Team, 2020). Predictors included counter calling or not, phase of interaction (Phase 1, Phase 2, Phase 3) and day of recording (Day 1 [control], Day 2 [playback]) (as defined above). Regressions on randomized data (N = 1,000) were conducted for a subset of these analyses (e.g., the counter calling analysis) to determine that these results were not spurious due to the clear limitation in sample size given an event of 1 (also see Supplemental Materials).

Results and discussion

Results shown in Fig. 3 indicate that substantial variation was found among the latencies in Twain’s calling behavior both across the playback phases (Figs. 3A and 3B) on Day 2 and between calls produced on Day 1 (control) vs Day 2 (playback) (Fig. 3C). For the latter, latencies between Twain’s calls were overall shorter on Day 2 than the calls recorded on Day 1, and all phases on Day 2 showed significantly shorter latencies to those on Day 1 (Fig. 3B). Notably for the former on Day 2, latencies between Twain’s calls were significantly shorter for Phase 1 during “Engagement” than either Phase 2 during “Agitation” or Phase 3 during “Disengagement”. Phase 2 latencies were also shorter than Phase 3 latencies albeit marginally so ( β = −0.30, p = 0.05). Day 1 latencies were significantly higher than all three phases on Day 2 (see Fig. 3B; Day 1-Phase 3: β = −0.30, p = 0.001). Finally, latencies between Twain’s calls were shorter when the call was in direct exchange with the exemplar than when it was not (across the entire time series regardless of phase; Fig. 3D). These results indicate that Twain was modifying her vocalization rate based upon the broadcast of the whup exemplar. Her motivational state in responding suggests either arousal, valence or some combination of both (Briefer, 2012).

Results shown in Fig. 4 indicate that, in addition to shorter latencies between Twain’s calls during interactive playback, matching between Twain’s vocalization rate and that of the exemplar was higher during Phase 1 than Phase 2 of the playback period. For matching that was categorized as a binary match or not (using the criterion indicated above), latencies between the exemplar and Twain’s calls produced in Phase 1 during “Engagement” were 9.9 times more likely to match the latencies of the previous exemplar than in Phase 2 during “Agitation” (Fig. 4A). Because only one E-T pair was recorded in Phase 3 as we had stopped broadcasting exemplars early on during this phase, we omitted this phase from the analyses. For the analysis in which we compared the absolute discrepancies in latency ratios during Phase 1 and Phase 2, we found a significantly lower degree of discrepancy in Phase 1 than Phase 2 (Fig. 4B). In Fig. 4C, we provide a pie-chart representation of the ratio discrepancy as a time series for visualization. (Note that the results for the T-E latency matching can be found in Supplemental Materials including a comparison between the two sets of latency-matching pair types- E-T and T-E pairs). These results indicate that a greater amount of vocal synchrony was occurring during Phase 1 than during Phase 2. Vocal and behavioral synchrony are associated with pair bonding, group cohesion, and even empathy in natural communicative exchanges between humans (Wood et al., 2021; Lord et al., 2015; Imel et al., 2014; Xiao et al., 2013; Bowling, Herbst & Fitch, 2013) and between nonhuman animals (Ravignani, 2018; Greenfield et al., 2021; Patel et al., 2009; King & McGregor, 2016; Hausberger et al., 2020; Demartsev et al., 2023; Herzing, 2015), again suggesting that Twain was actively engaged in the turn-taking exchange with the broadcasted exemplar.

Figure 4 Latency matching between Twain’s calls and exemplars.

(A) Logistic regression indicating a greater proportion of matching between Twain’s calls and the exemplar was found (9.9x more likely) in Phase 1 during “Engagement” than in Phase 2 during “Agitation”. (B) Negative binomial regression on the difference or discrepancies between the latency ratios of Twain’s calls and the exemplars where discrepancy was significantly lower for Phase 1 than Phase 2. (C) Visualization (pie charts) of the latency ratio among successive pairs of Twain’s calls and exemplars as a time series. Red line separates Phase 1 from Phase 2.

In sum then, the data on latencies between Twain’s calls and the latency matching between Twain’s calls and the exemplars shown in Figs. 3 and 4, in relationship to the independent data collected on Twain’s surface and respiratory behavior, strongly indicate that Twain was actively engaged in a type of vocal coordination (antisynchrony), and specifically as a “coupled oscillator” with our playback system (Demartsev et al., 2018) (especially during Phase 1 of the exchange). This finding suggests that Twain was aware of our exemplar’s temporal dimension and that she had the capacity to coordinate in an antisynchronous counter calling framework. Her concurrent behavioral data suggest that she was also exhibiting changes to both arousal and valence during the encounter: certainly excitement and possibly the onset of agitation during Phase 2. However, she was not sufficiently aroused that she initiated calling in bouts or produced additional call types, which has been demonstrated previously (Dunlop, 2017). Interestingly, one reason that might explain the establishment of Phase 1 engagement during the playback with Twain is that the whup call used as the exemplar was serendipitously and opportunistically recorded on the previous day during a control period (Day 1). This call was recorded from one of the group of animals present during that recording, which included Twain and eight other whales in a dispersed aggregation (see Supplemental Materials for details). We can only speculate that this initial strong response to our whup call exemplar was due to either a mirroring effect of playing back Twain’s own call to herself or increased interest due to the fact that the call was identifiable to one of her group members. Although perhaps unlikely, the former would suggest a highly evolved cognitive capacity of self-recognition in humpback whales that could be further explored using a type of interactive adaptive design (see “Lessons Learned: Directions for Future Research” below). If the latter, the initial and then eventual waning of engagement found across the playback exchange provides us with deeper insights into how we can improve future playback designs for unraveling vocal structure and meaning in animal communication. For example, one promising area that this study points to is discerning the salient turn-taking rhythm of each species’ communication system (Healy et al., 2013), which in humpbacks can be evaluated by investigating how the diverse social sounds that humpback whales produce are binned by individuals into rhythmic units, like their song appears to be structured (Schneider & Mercado, 2019). We describe further insights in the final section below.

Lessons learned: directions for future research

In all, from this rare opportunistic event and the accompanying acoustic analyses, we can suggest several important insights for future studies attempting to solicit prolonged acoustic interactions between humans and nonhuman species, such as humpback whales.

First, a library of salient well-processed stimuli should be prepared and include stimuli recorded both outside of and within the current field season (Deecke, 2007). For example, our capture of the exemplar on Day 1 from a group of animals that included Twain, while serendipitous, suggests that calls recorded from animals known to the interacting individual are more salient than randomly selected calls (McCowan & Newman, 2000; King, 2015; King & McGregor, 2016). In our study, playback trials conducted on multiple previous and subsequent days using whup exemplars from archived recordings (N = 15) did not solicit physical or acoustic interaction. Furthermore, the salience of playback stimuli may depend on the rate of change for species-specific signals where recent variants may be more salient for signals that are being modified at a faster rate. For example, a playback experiment conducted with Campbell’s monkeys found that individuals responded more often to current variants of a signal compared with older versions of the same signal (Lemasson, Hausberger & Zuberbühler, 2005).

Second, appropriate equipment to record and broadcast stimuli during playback experiments with high fidelity is certainly desirable. We acknowledge that the quality of our playback suffered from a partial low-end frequency drop out by the Lubell LL916 (frequency response: 200 Hz−23 kHz; see Supplemental Materials for additional details). This is a common problem in conducting playback studies with other species whose acoustic range is different from humans (e.g., infrasound and ultrasound). Generally, most speakers are designed for human hearing and even those that are designed to be used for specialized playbacks are still limited in the dynamic range of their frequency responses (Cetacean Research Technology, personal communication). One solution is to use multiple speakers covering the required ranges that can be selectively triggered based upon the acoustic features of the playback stimuli being broadcast. This should be recommended, however, with the caveat that even with an appropriate frequency response, the recording and playback of any stimuli will still be to some extent modified simply because of the act of recording and broadcasting via human-made equipment (even when broadcasting to our own species). Indeed, one example of this reflected in our system is a mismatch between recorded and ambient background noise (see Supplemental Materials for spectrograms). Some researchers opt to use automated algorithms to reduce the noise in such recorded signals but acknowledge that such filtration can distort the original signal, especially if the signal is embedded in such noise (as in our case). More generally, we note that sound production even by live individuals can be distorted/modified during transmission by variation in water (or air) density, temperature/thermocline, current, topography/bathymetry, and ambient noise conditions, to which receivers must adapt. Therefore, all said, our system, although technically compromised by these issues, still managed to solicit an extended exchange/interaction with Twain indicating that the components retained and broadcast in the playback were salient enough to elicit a response. Nevertheless, diagnostic tests prior to playback experiments should be conducted to ensure that recorded playback stimuli retain as high a fidelity as possible to the original acoustic components (with particular attention to features known to be salient to the study species) and cover the frequency (or amplitude) of the study animal’s natural range with minimal noise.

Third, we recommend that multiple hydrophones (or microphones) be used and deployed at different depths (locations) to capture the configuration of acoustic interactions. The data from each specific hydrophone’s recording did not accurately capture the full interaction event. Indeed, the data from the starboard-side hydrophone revealed additional responses by Twain not detected by the port-side hydrophone, which was deployed adjacent to the speaker. These differences in the detection of sounds from these two hydrophones could be due to differences in signal sensitivity, location of deployment on the vessel, and other behavioral factors of the subject including distance from or orientation to each hydrophone. Deploying multiple hydrophones (an array would be ideal) provides additional information with which to corroborate acoustic data and/or behavioral observations.

Finally, we want to stress the value of collecting data on multiple modalities of behavior in the pursuit of understanding the function and flexibility of animal signals. Many studies on animal communication seek to understand signal function by characterizing the context in which the signal is solicited. Independent or behavioral observations that are blind to experimental procedures are particularly useful in providing contextual clues to acoustic behavior. These types of behavioral clues allowed our study team to break up our encounter into different phases based on the behavior of the targeted recipient of the experiment. Identifying these behavioral changes can make adaptive playback design more engaging and salient for experimental targets as well as provide contextual clues for scientists focused on analyzing the function of animal signals. Although the use of a handheld video recorder was sufficient for our group to record behavioral observations on the top-deck of the vessel, we would suggest using cameras with a wider lens (such as a 360 ∘ camera or drone) to facilitate observations over a more expansive area. Drones are particularly promising for documenting these types of interactions because they can follow the animals and provide a closer, subsurface view with more detailed account of behavioral sequences and interactions. Wide angle underwater cameras such as Go Pros would also be useful in the case of aquatic animals to capture the underwater activities of the targeted individual(s).

More importantly, we suggest that an “interactive” approach to playback where the broadcasting of the stimuli should be modified in response to the animals’ acoustic or behavioral response during the playback period, including but not limited to the timing of playbacks. While our results certainly indicate the importance of dynamical control over the temporal element in a playback, a major limitation in our playback study was the inability to modify anything other than the timing of the playback. This limitation may have either frustrated or disinterested the whale over the course of the three phases as indicated by her surface behavior. This type of more complex manipulation can likely be accomplished with readily available software such as Audacity (http://audacityteam.org/) or Ableton (https://www.ableton.com/en/) and sufficient computer resources to process recorded signals in almost real-time. A manually-operated networked computer system consisting of three computers: one to record responses, a second to manipulate recorded responses and a third to playback the modified sounds would be ideal to conduct adaptive playback. Such an approach would permit real-time recording, manipulation, and subsequent playback of calls to be conducted, allowing modification of both temporal and spectral features that can elicit differential responses in real time.

We therefore conclude that although considerably challenging, adaptive interactive acoustic playback is possible to conduct given recent innovations in technology. Such technology allows immediate and adaptable manipulation of playback signals as a function of the study subject’s current and changing behavior (King, 2015; King & McGregor, 2016; Butkowski et al., 2011; McGregor, 2000; McGregor et al., 1992; Douglas & Mennill, 2010; Miller et al., 2009; Nielsen & Vehrencamp, 1995; Burt, Campbell & Beecher, 2001). Such an adaptive playback design incorporates dynamics into this well-established experimental framework and allows previously intractable questions to be addressed about signal function and meaning by probing the system and its subjects in real-time. As such, this approach offers a more powerful tool for examining animal communication than traditional passive approaches by enabling the exploration of the interactive and dynamical features of communicative behavior.

Supplemental Information

Supplemental Information 1 Equipment Specifications.

Click here for additional data file.

We thank Marc Choquette (Captain, R/V Glacier Seal, Juneau), Don and Denise Bermant (M/Y Blue Pearl, Vancouver), Tony Gilbert (Program Director, Seakeepers), Collette Costa, and Debbie Kolyer for logistical support. We also thank Clark Snodgrass, Katie Zacarian, Aza Raskin and Britt Selvitelle for support in observational data collection and whale identification, and James Crutchfield for his participation in the playback trial as well as initial discussions during the 2021 field season. The opinions expressed in this publication do not necessarily reflect the views of Templeton World Charity Foundation, Inc.

Additional Information and Declarations

Competing Interests

Author Contributions

Animal Ethics

Data Availability

Lisa Walker is the director of the Grooved Whale Project. Jodi Frediani Photography is a business owned by Jodi Frediani. Neither has competing interests.

Brenda McCowan conceived and designed the experiments, performed the experiments, analyzed the data, prepared figures and/or tables, authored or reviewed drafts of the article, and approved the final draft.

Josephine Hubbard conceived and designed the experiments, performed the experiments, authored or reviewed drafts of the article, and approved the final draft.

Lisa Walker conceived and designed the experiments, performed the experiments, analyzed the data, prepared figures and/or tables, authored or reviewed drafts of the article, and approved the final draft.

Fred Sharpe conceived and designed the experiments, performed the experiments, authored or reviewed drafts of the article, and approved the final draft.

Jodi Frediani conceived and designed the experiments, performed the experiments, authored or reviewed drafts of the article, and approved the final draft.

Laurance Doyle conceived and designed the experiments, authored or reviewed drafts of the article, and approved the final draft.

The following information was supplied relating to ethical approvals (i.e., approving body and any reference numbers):

National Marine Fisheries Service

The following information was supplied regarding data availability:

The raw data are available at Dryad: McCowan, Brenda et al. (2023). Data from: Interactive bioacoustic playback as a tool for detecting and exploring nonhuman intelligence: “Conversing” with an Alaskan humpback whale [Dataset]. Dryad. https://doi.org/10.5061/dryad.ht76hdrn0.

The timeline of interaction, latency data, latency matching data, amplitude data, randomizations conducted as shown in Fig. 3E, specifications of equipment used and both wav files and spectrograms of some of the whup calls, and videos synced with hydrophones of the encounter are available at Zenodo: McCowan, B., Hubbard, J., Walker, L., Sharpe, F., Frediani, J., & Doyle, L. (2023). Data from: Interactive bioacoustic playback as a tool for detecting and exploring nonhuman intelligence: "Conversing" with an Alaskan humpback whale. Zenodo. https://doi.org/10.5281/zenodo.8247515.

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
