# Peer review of "Interactive bioacoustic playback as a tool for detecting and exploring nonhuman intelligence: “conversing” with an Alaskan humpback whale"

_PeerJ, doi:10.7717/peerj.16349_

## Round 0.1 · original submission · Major Revisions

Please address all comments of reviewers 1 and 2 (please note attached separate review provided by reviewer 2).

·

Basic reporting

see section 4 for full review

Experimental design

see section 4 for full review

Validity of the findings

see section 4 for full review

Additional comments

Thank you for the opportunity to read and review your work and I apologize for it taking a bit longer than expected. I think that interactive playbacks are an understudied tool and wish that more studies would attempt this design.

I fully agree with your claims in LL84-119 that vocal interactions rely not only on the signals themselves but also on the “correct” temporal structure of the interaction. This is an important factor to account for when studying vocal communication and I am well aware of the methodological and analytical difficulties involved. So, in the general sense, I see great value in this work as it demonstrates that these experiments are doable in field settings.
I do however have a few comments and questions which I hope you can address to better clarify the design of the study and its context.

1. The first and main issue is the design of the playback and the defined experimental conditions (L192-194):
I don't think I fully understand what "spontaneously derived intervals" mean. Were the intervals between the exemplar calls somehow contingent on the response of the whale? By interspersed do you mean that an exemplar call was repeatedly played after the wales vocal response was detected? Or the calls were just randomly spaced throughout the playback track and happen to intersperse with some of the responses?
How the playback activation conditions were defined? You mention that the first response came after 3 playback calls. So at least these first calls were likely played with some random interval between them.
These details are important to understand if the experiment is a true “interactive” playback with the stimulus being dynamically adjusted to the animals’ responses. If the playback activation was set randomly it is not a fully interactive playback design, at least not according to the definitions of King, S. L. (2015).

2. Acoustic analysis of calls:
In L227 you mention that calls recorded on day 1 were acoustically analyzed. It is not mentioned what analysis was performed and what “considerable investigations” led to a decision to focus only on latency. I agree that the possibilities for manipulations of the playback track were very limited with only 1 call exemplar. If the quality of recorded calls permits, an acoustic analysis could have helped confirm caller identity (see my next comment) or give additional metrics for comparing arousal (Briefer et.al, 2015). Again, I am well aware that acoustic analysis with 6 calls recorded in field settings can be of very limited use. But still feel that a more substantial justification should be given for completely excluding it from the results.

3. Latency between Day1 and Day2:
You show a decreased latency between day 1 and day 2 calls (fig3B). How clear was the identity of the caller on day 1? In L171 you mention that the calls were recorded from an aggregation of 9 whales and as I understood from LL302-304 you are not sure about the identity of the emitter. How certain are you that all 6 calls on day 1 were produced by the same animal? If so, were there any vocal responses from the whales present in the vicinity?
If no other animal responded to naturally emitted calls on day 1 do you have an explanation for how the playback on day 2 could have been different to stimulate such a strong response?

4. E-T matching:
By latency matching do you mean the matching of what response times? Were you comparing the times of consecutive E-T pairs (the animal responding to the stimulus after a more or less constant lag time)? Or were you comparing the matching between consecutive T-E and E-T pairs (the animal is matching the response time to the response time of its interlocutor)? Looking at the pie charts I assume the latter but still not 100 sure as it is not stated very clearly in the methods
The latter option would support the idea of entrainment or oscillator dynamics to maintain turn-taking. The former would be harder to explain. Overall shorter latencies could lead to closer matches without the involvement of any coordination mechanism.

5. Data availability:
I appreciate your provided supp data and r code. I think that this dataset would be easier to interpret with two small additions: 1) more detailed explanations about the info provided on each of the spreadsheets. Perhaps also explaining the column headings. For example, I am not sure what cc means (in Twain Latency Data tab). Additionally, a full, raw, timeline of the interaction would be nice. Not only the calculated call latencies but the whole sequence of playback and Twain calls with real time-stamps. 2) could you add a more detailed comment on the Rscript explaining what analysis this code is related to? I figured eventually that it is Fig3E but it could save the readers some time.

6. Methods:
Can you give a slightly more substantial explanation of why you considered the recorded exemplar as high quality? Which measurements were done on the recorded calls? A high res spectrogram or audio samples of the calls as a supplement can help to better understand the quality of the exemplars required for playbacks.

Additionally, in the Methods chapter you present some post hoc analyses and results (L198, L209). Those might be more suitable for the results chapter.

7. Introduction:
In LL47-50 you talk about parallels between human and whale songs and mention similar frequency bands, tonal and broadband calls as well as the variable duration of vocalization. I feel that those features are not unique to whales and exist in multiple animal species, mammals and birds for sure. So, it seems to me that focusing on whales for this parallel is not fully justifiable. Also, I don’t fully agree with the statement in L58. Mainly since I don't quite get the reasoning of why a diversity of signals is required for interacting with "non-human" intelligence.
I also do not see this as a "unique" opportunity as playback procedures are routinely performed in various animal systems. I see the interactive playback design as a unique part of this study not the choice of the specific study system.

LL123-126: I completely agree with these points but think that they are beyond the scope of the paper. You discuss inter-species communication however I feel that the playback experiments do not fully qualify for that. You have sophistically simulated a dynamic intra-species communicational event. The signal that was received by the whale I not foreign and as you point out in L208-209 you consider the response behaviour to be “an exchange” with the playback signals and not “a causal human-animal interaction”.

8. Lessons learned:
L320 you mention an interesting point, regarding the lack of response to “old recordings”. I wonder how many such trials were attempted? How certain are you that the lack of response is attributed to the “age” of the recordings/the identity of the caller and not to the informational content of the signal?

L329: You mention “low-end frequency drop out and a mismatch between the recorded and ambient background noise” I could not see where in the supplement this was mentioned. I am sorry if I missed it. In any case, as mentioned earlier, more details on the provided datasets would be helpful.
L371 you mention technologies allowing adaptive/interactive playbacks. I feel that given the novelty of this experimental design, more specific details on the procedures and equipment allowing interactive playbacks would be extremely useful. For example, details on the method of playback activation (manual vs automatically controlled). Determining the relevant activation lag (response time varies between species as you mentioned in L96).

I hope that you will find my comments useful and looking forward to seeing this paper published!

Reviewer 2 ·

Basic reporting

No comment

Experimental design

No comment

Validity of the findings

No comment

Additional comments

Please find my comments in the attached PDF.

Annotated reviews are not available for download in order to protect the identity of reviewers who chose to remain anonymous.

---

## Round 0.2 · Minor Revisions

You need to consider the comment of Reviewer 1 that the speaker model and its limitations are clearly provided. In fact I would like to ask the authors to have this limitations of the speaker checked and discussed in the Discussion section.

Reviewer 2 ·

Basic reporting

no comment

Experimental design

no comment

Validity of the findings

no comment

Additional comments

All of my comments have now been addressed.

As a final note regarding the type of speaker used for the experiments: I have now been able to see the equipment specifications. First, the exact model should be given in the excel supplementary document (and not only 'Lubell speaker'). Second, and most critically, this speaker's frequency response is 200Hz-23kHz which means that a whole part of the signal is completely overlooked/non-represented in the stimuli used for broadcast (the initial low-frequency growl with a 56-187 Hz frequency range, see Wild & Gabriele, 2014: 'Putative contact calls made by humpback whales (Megaptera novaeangliae) in southeastern Alaska').
Although indeed this was still enough to elicit a response (as stated in the manuscript) I would highly advise against using this equipment in any further trial if meaningful interpretations are to be drawn.

---

## Round 0.3 · accepted · Accept

All comments have been adequately addressed; the manuscript is now ready for publication.